# Optimal Design of a Plenum Fan with Three-Dimensional Blades

**Kyung Jung Lee** [1], **Il Wung Park** [1]**, Ki Suk Bang** [1]**, Yeong Min Kim** [1] **and Young Chull Ahn** [2,*]

[1]   LG Electronics, Changwon 2nd Factory, Seonsan-dong, Changwon-city, Keongsang Namdo 51554, Korea;
     kyungjung.lee@lge.com (K.J.L.); ilwung.park@lge.com (I.W.P.); gojang77.bang@lge.com (K.S.B.);
     Yeong.kim@lge.com (Y.M.K.)
[2]   Department of Architectural Engineering, Pusan National University, Busan 46241, Korea
*    Correspondence: ycahn@pusan.ac.kr

**Abstract:** We successfully designed an optimized plenum fan with a three-dimensional, smooth, curved blade. The optimized model revealed that the static pressure in the channel had been increased uniformly and stably, and the flow separation at the leading edge was significantly reduced. According to simulations, the three-dimensional blade stabilized the fluid flow, and the flow friction was reduced by suppressing the flow separation as much as possible so that both the static pressure and the static efficiency were clearly improved in comparison with those of the original model. As a result, the static efficiency was improved by 6.3% compared with that of the original model.

**Keywords:** plenum fan; blade profile; static pressure; static efficiency; velocity distribution; pressure distribution

## 1. Introduction

Centrifugal fans are commonly used for industrial purposes or air conditioning because they are more efficient and quieter than other fans at the same revolutions per minute (rpm). Specifically, the plenum fan is quiet and small, produces excellent airflow compared with existing centrifugal fans since it has no housing, and is widely used for air conditioning and ventilation.

In industrial use or air conditioning, a fan's air-conveying power uses the majority of the consumed energy. Therefore, to reduce energy consumption and promote efficient use, the most important factor to improve is the fan efficiency. The blade, as the impeller of the centrifugal fan, is the most crucial part in determining the performance of the fan. The blade's geometry is an important design consideration for the flow separation at the blade surface and the stabilization of the flow pattern. Various studies have been carried out to improve the blade's function because it plays a decisive role in the overall performance, such as the internal flow and efficiency of the fan.

Wu et al. [1] proposed an optimal profile design method for centrifugal impeller blades by controlling their velocity distribution. Dou et al. [2] carried out numerical simulations on the flow of a plenum fan equipped with rotating vaneless diffusers with different diameter ratios, and proposed the optimum diameter ratio according to the flow coefficient. Lee et al. [3] analyzed the effects of the bending length ($\ell$/c) and the bending angle ($\theta$) in the leading-edge direction on the impeller trailing edge of the centrifugal fan as design variables by numerical simulation. Park et al. [4] reviewed and studied the application of an airfoil impeller to improve the aerodynamic performance of a centrifugal fan in high-speed rotation. It was confirmed that the application of an airfoil impeller greatly reduced the flow separation that occurs on the blade pressure and diffuser pressure surfaces compared with the existing blade shape. Siwek et al. [5] presented a numerical model of the centrifugal fan and conducted experiments to verify the fan's performance characteristics. As a result, the accuracy of the

numerical model was verified. Kim et al. [6] presented a numerical analysis model for a splitter-type centrifugal fan and confirmed that the splitter blade improved the fan's overall performance. Chunxi et al. [7] conducted a study on the performance of centrifugal fans with 5% and 10% increases in the impeller size by extending the tip of the blade without changing the size of the volute. By extending the blade tip to increase the size of the impeller, the flow rate, voltage, and axial force of the fan increased during operation. However, although the nonuniformity of the flow in the volute increased, the overall efficiency of the fan decreased. Lin et al. [8] conducted a numerical analysis for flow visualization, torque calculation, efficiency, and noise for centrifugal fans with a small-diameter rear wing for computer cooling. Ni et al. [9] conducted a numerical investigation using ANSYS Fluent on the internal flow of a Sirocco fan to investigate the effects of the inclination angle of the blades on the fan performance. The effects of the inclined blade were demonstrated by the variations in static pressure, efficiency, and pressure and velocity distributions at various inclination angles. Additionally, there have been numerical analyses and experimental studies on the method of reducing noise by changing the structure of the volute or the material of the volute tongue of the centrifugal blower [10–14]. Most of the previous studies focused on the improvement of the centrifugal fan with a scroll. In some scroll-less plenum fan studies, the airfoil had a simple two-dimensional shape, so the peak efficiency was relatively low at about 70%.

In this study, we optimized the airfoil blade of the plenum fan and compared its performance with that of the original model. Our optimally designed blade had four layers, and the iteration method was introduced to calculate the optimal design value of each layer.

## 2. Numerical Investigation

### 2.1. Numerical Methods

The commercial computational fluid dynamics (CFD) solver of ANSYS-CFX 17 [15] was used for the numerical analysis to evaluate the performance of the plenum fan. Calculations were performed on a Windows 10 computer (Intel (R), 3.2 GHz, 64 GB RAM) with a 64-bit operating system. For stability of the analysis results, convergence criteria were set to $1 \times 10^{-5}$.

The fan's inlet and outlet were set to an open boundary condition, the inlet pressure was set to atmospheric pressure, and the wall condition was set to a no-slip wall condition. The impeller's rotation speed was set to 1100 rpm, and the simulation was carried out based on the flow rate given in the product. The numerical calculations were discretized using three-dimensional, incompressible Reynolds-averaged Navier–Stokes (RANS) equations. The normal numerical analysis was carried out using the pressure based on the fully coupled implicit method. The turbulence model used a shear-stress transport (SST) model, which is useful for analysis of the flow separation.

Generally, in flow analysis using CFD, attention should be paid to the y+ of the first grid point to analyze the boundary layer of the wall. y+ should also be 1–2 or less in the case of a low-Reynolds-number turbulence model when analyzing from the viscous sublayer of the wall. However, to satisfy these conditions, many grids packed densely on the wall surface are required, significantly increasing the total number of grids. To solve this, a wall function is usually used to reduce the number of grids concentrated on the wall. With the automatic wall function of CFX, if the first grid y+ is located between 1 and 100, the flow analysis results and the theoretical equations agree well with the grid y+ such that the y+ problem on the wall can be solved [16]. When analyzed using the automatic wall function in the CFX, the flow analysis can be performed without worrying about the y+ by simply driving the grid on the wall such that the optimization results based on the flow analysis can be trusted [17].

### 2.2. Governing Equations

The continuity equation and momentum equation applied in this study can be expressed in a conservation form as follows:

Continuity equation:

$$\frac{\partial u_i}{\partial x_i} = 0 \tag{1}$$

where $u_i$ is the instantaneous velocity in the i direction.

Momentum equation:

$$\frac{\partial}{\partial t}(\rho u_i) + \frac{\partial}{\partial x_j}(\rho u_i u_j) = -\frac{\partial P}{\partial x_j} + \frac{\partial \tau_{ij}}{\partial x_j} \rho f_i \tag{2}$$

where $P$ is the static pressure, $\tau_{ij}$ is the viscous stress tensor, and $f_i$ is the body force.

In Newtonian fluids, $\tau_{ij}$ can be expressed in terms of the velocity gradients, as shown in Equation (3).

$$\tau_{ij} = \mu\left(\frac{\partial u_i}{\partial x_j} + \frac{\partial u_j}{\partial x_i}\right) - \frac{2}{3}\mu\left(\frac{\partial u_m}{\partial x_m}\right)\delta_{ij} \tag{3}$$

where $\mu$ is the fluid dynamic viscosity and $\delta_{ij}$ is the Kronecker delta.

Equations (2) and (3) can be used to obtain the Navier–Stokes equation (Equation (4)):

$$\begin{aligned}
&\frac{\partial}{\partial t}(\rho u_i) + \frac{\partial}{\partial x_j}(\rho u_i u_j) \\
&= -\frac{\partial P}{\partial x_j} + \frac{\partial}{\partial x_j}\left(\mu\left(\frac{\partial u_i}{\partial x_j} + \frac{\partial u_j}{\partial x_i}\right) - \frac{2}{3}\mu\left(\frac{\partial u_m}{\partial x_m}\right)\delta_{ij}\right) + \rho f_i
\end{aligned} \tag{4}$$

To simplify the problem, we used the method of averaging the Navier–Stokes equation. Equation (5) shows the RANS equation:

$$\begin{aligned}
&\frac{\partial}{\partial t}(\overline{\rho u}_i) + \frac{\partial}{\partial x_j}(\overline{\rho u}_i \overline{u}_j) \\
&= -\frac{\partial \overline{P}}{\partial x_j} + \frac{\partial}{\partial x_j}\left(\overline{\mu}\left(\frac{\partial \overline{u}_i}{\partial x_j} + \frac{\partial \overline{u}_j}{\partial x_i}\right) - \frac{2}{3}\overline{\mu}\left(\frac{\partial \overline{u}_m}{\partial x_m}\right)\delta_{ij}\right) + \frac{\partial}{\partial x_j}\left(-\overline{\rho u_i' u_j'}\right) \\
&+ \overline{\rho f_i}
\end{aligned} \tag{5}$$

### 2.3. Geometrical Model and Mesh

The original plenum fan model PRL-560L0(LG Electronics, Korea) was used in this study. The geometric shapes are shown in Figure 1, and the specifications are shown in Table 1. Figure 2 shows the computational domain. The radius of the computational domain was three times the diameter of the plenum fan, and the height was set to 30 times the width of the plenum fan. In the figure, Block 1 is the area at which the flow enters through the bell mouse, Block 2 is the fan part and the rotating flow area, and Block 3 is the area where there is rotational flow. The flow field was treated as a wall in this work. As a boundary condition, the inlet condition was the pressure inlet condition, the outlet condition was the mass flow rate condition, and the air density corresponded to 25 °C. The size of the domain was considered large enough to avoid interference in the flow.

**Table 1.** Rated variables of the plenum fan.

| Variables | Units | Value |
|---|---|---|
| Flow coefficient | - | 0.675 |
| Static pressure | Pa | 392 |
| Static pressure efficiency | % | 71.8 |
| Impeller diameter | mm | 640 |
| Impeller outlet breadth | mm | 167 |
| Number of blades | - | 7 |
| Revolving speed | rpm | 1100 |

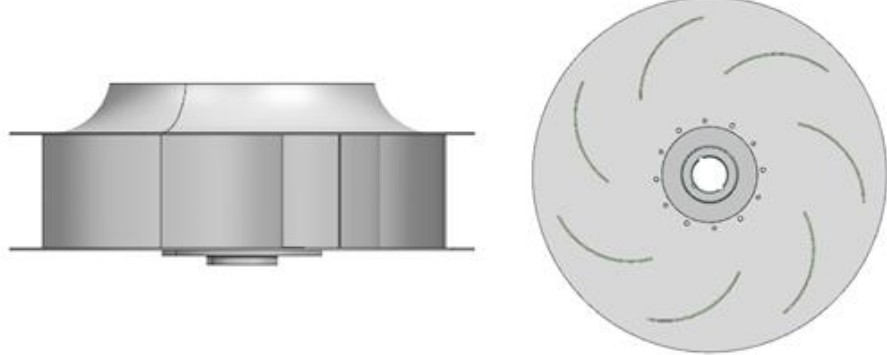

**Figure 1.** Geometric shape of the original model (PRL-560L0).

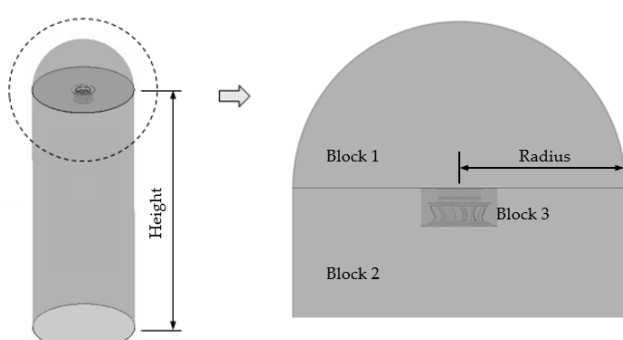

**Figure 2.** Computational domain.

Figure 3 shows the axial cross section of the original model's mesh system. Since the impeller and shrouds were curved and exhibited complex geometrical shapes, a high-quality structured grating could not be applied, so an unstructured grating was used. For analysis, 9.4 million gratings were constructed for the impeller and the bell mouth, and 7.4 million tetrahedra and 2 million wedges were applied.

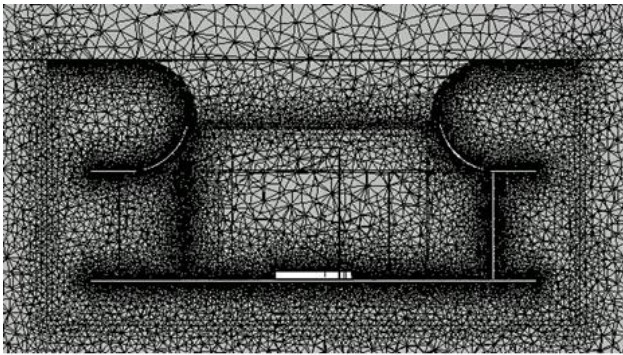

**Figure 3.** Mesh system.

*2.4. Experimental Results and Comparison with the Numerical Simulation*

To examine the validity of the original model's numerical results, they were compared with the experimental results under the design conditions. The equipment used in the experiment was a laminar flow multi-nozzle chamber that satisfied Figure 15 of ANSI/AMCA standard 210 [18] and ANSI/ASHRAE standard 51 [19]. Figure 4a shows the schematic diagram of the chamber, and Figure 4b shows the plenum fan test scene.

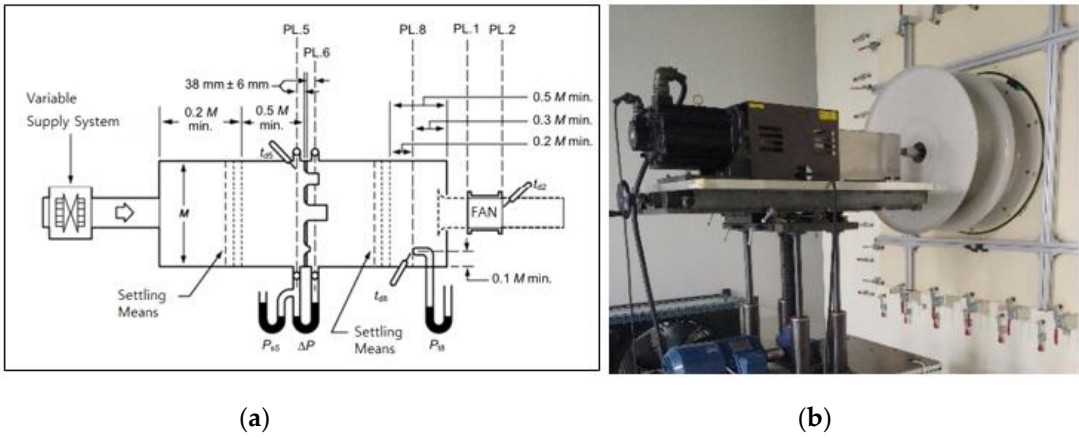

(a)                                                                                      (b)

**Figure 4.** (**a**) Schematic diagram of multi-nozzle chamber; (**b**) Test scene.

Figure 5 shows that the CFD results and the experimental results are in good agreement with the rated airflow rate and the static pressure conditions.

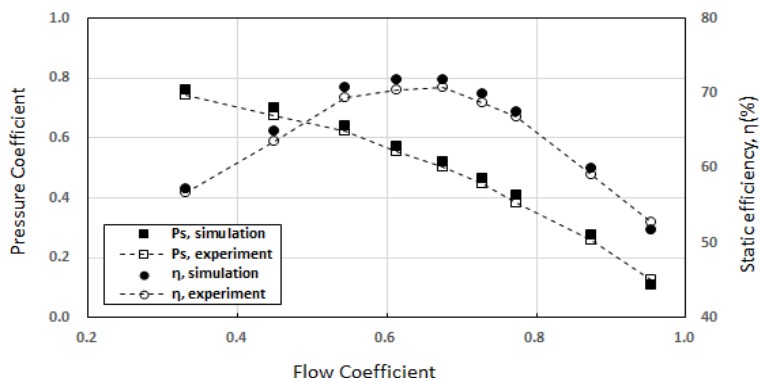

**Figure 5.** Comparison between the numerical and experimental results (at 1100 rpm).

## 3. Optimization of the Blade Profile

In this study, to design the optimum airfoil of the plenum fan, the blade was divided into four layers, and the optimum airfoil was designed for each layer. Figure 6 shows a three-dimensional curved shape and an image of the blade with four layers.

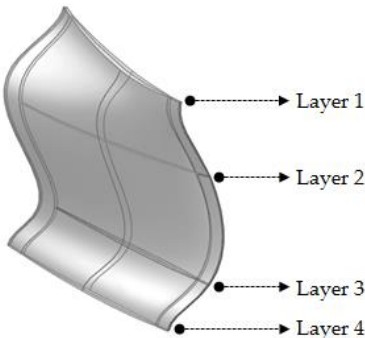

**Figure 6.** Layer definition of the plenum fan blade.

The airfoil was designed using NACA (National Advisory Committee for Aeronautics) four-digit. The basic design parameters required to design the airfoil are shown in Figure 7. $D_1$ and $D_2$ are the values that determine the radial positions of the leading and trailing edges, and can be defined as the

inner and outer diameters. $\gamma$, $\alpha$, and $\beta_1$ are all angles, where $\gamma$ is the relative position of the airfoil's leading edge, $\alpha$ is the code angle corresponding to the airfoil length, and $\beta_1$ is the incidence angle. Note that when defining $\beta_1$, the minimum value of $\beta_1$ ($\beta_{1,min}$) is determined using $D_1$, $D_2$, and $\alpha$. This value can be calculated using the following equations:

$$c = \sqrt{\left(\frac{D_1}{2}\right)^2 + \left(\frac{D_2}{2}\right)^2 - 2\frac{D_1}{2}\frac{D_2}{2}\cos(\alpha)} \tag{6}$$

$$2\frac{D_1}{2}c\cos(\beta_{1,min} + 90) = \left(\frac{D_1}{2}\right)^2 + l_c^2 - \left(\frac{D_2}{2}\right)^2 \tag{7}$$

where $c$ is the code length.

$\theta_1$, $\theta_2$, and $\varphi$ are defined using the geometric relationship as follows:

$$\varphi = \cos^{-1}\left[\frac{C^2 + \left(\frac{D_2}{2}\right)^2 - \left(\frac{D_1}{2}\right)^2}{2c\left(\frac{D_2}{2}\right)}\right], \theta_1 = \alpha + \varphi - \left(\frac{\pi}{2} - \beta_1\right), \theta_2 = \frac{\pi}{2} - \beta_2 - \varphi \tag{8}$$

The thickness distribution for the NACA four-digit section was selected to correspond closely to that for the wing sections and is given by:

$$y_t = \frac{t}{0.2}c\left[0.2969\sqrt{\frac{x}{c}} - 0.1260\left(\frac{x}{c}\right) - 0.3516\left(\frac{x}{c}\right)^2 + 0.2843\left(\frac{x}{c}\right)^3 - 0.1015\left(\frac{x}{c}\right)^4\right] \tag{9}$$

where c is the code length, $x$ is a position along the code from 0, y is half the thickness at the $x$ position, and $t$ is the maximum thickness.

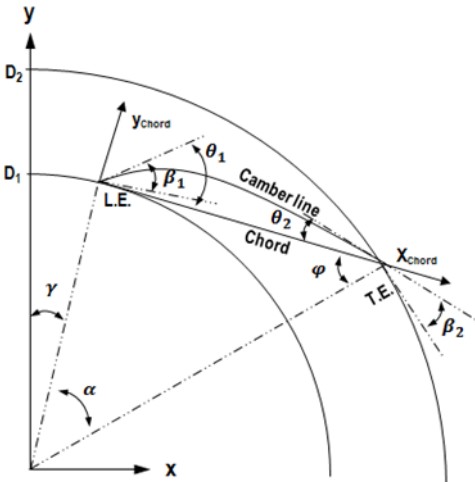

**Figure 7.** Airfoil design parameters.

In this study, the iteration method was introduced to optimally design the plenum fan with a three-dimensional blade. The flowchart of the iteration method used in the calculation is shown in Figure 8. The Taguchi method [20,21] was used for initial factor selection as a powerful statistical tool for designing an experiment utilizing orthogonal array tables. Orthogonal arrays that depend on the number of factors and levels are used to study parameter space. Numerous input variables can be assessed to determine their contribution to the output response variable using a fewer number of experimental runs. In this study, an experimental plan was established using the Taguchi method to select the initial factor values for each layer. Experimental plans were established using L12 ($2^{11}$) orthogonal arrays in Layers 1 and 4, and L16 ($2^{12}$) orthogonal arrays in Layers 2 and 3.

After selecting the initial values using the Taguchi method, the response surface method (RSM) was used to optimize the variables selected as the main factors through main-effect analysis. In using the RSM, the dependent variable to be optimized was selected efficiently.

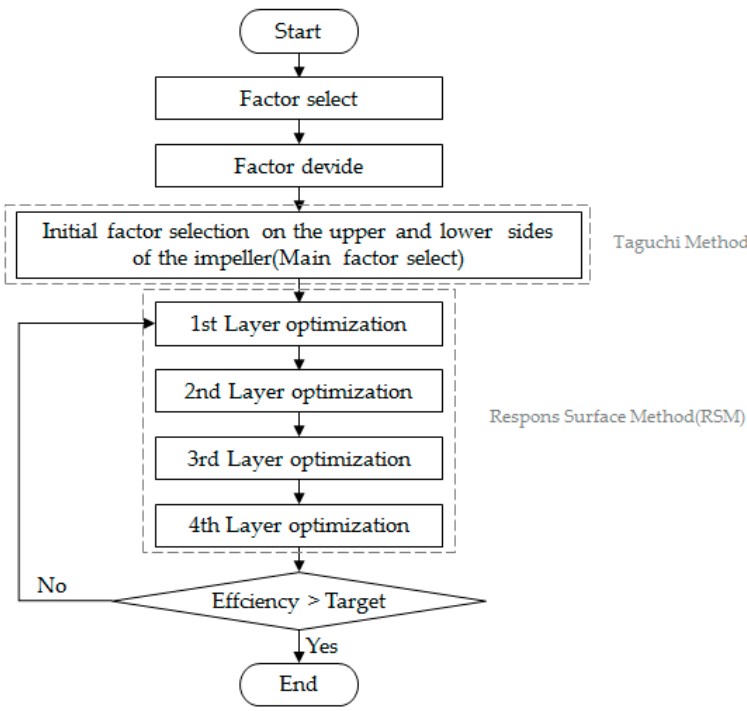

**Figure 8.** Flowchart of the iteration method for blade optimization.

The parameters of Layers 1, 4, and 2, 3 were simultaneously considered to attain the initial values for each layer. Layer 1 was selected as the plane where the blade met the shroud, and consisted of six design variables: $\alpha$, $\gamma$, $\beta_1$, $D_1$, $D_2$, and height. Layer 4 was selected as the plane where the blade met the back plate. It consisted of five design variables: $\alpha$, $\gamma$, $\beta_1$, $D_1$, and $D_2$. To select the major design variables, L12 orthogonal arrays were used for 11 factors, and screening analysis was performed by computational simulation. The main-effect analysis results of Layer 1 and Layer 4 are shown in Figure 9a, and the effects of $\alpha$ at Layer 1 and $D_1$, $D_2$, and $\gamma$ at Layer 4 were significant.

The RSM was applied to the first optimization in Layers 1 and 4. Four factors that showed a significant effect by main effect analysis were selected as factors of the RSM, and each factor was divided into three levels. Table 2 shows each factor and level. In the design of the experiment, central composite design (CCD) [22] was applied.

In Layers 2 and 3, we selected an arbitrary cross section of the blade, and these two layers consisted of six design variables such as $\alpha$, $\gamma$, $\beta_1$, $D_1$, $D_2$, and height. To select the major design variables, L16 orthogonal arrays were used for 12 factors, and screening analysis was performed by computational simulation. The main-effect analysis results of Layer 2 and Layer 3 are shown in Figure 9b, and the effects of $\gamma$ at Layer 2 and $\gamma$, $\beta_1$, and $D_2$ at Layer 3 were significant.

The second optimization in Layers 2 and 3 was performed by applying the RSM in the same way as the first optimization. After determining the initial shapes of Layers 2 and 3, four factors that showed significant effects in the main effect analysis were selected as factors of the RSM, and each factor was divided into three levels. The values of Layers 1 and 4 were used to optimize the results from the first optimization. Table 2 shows each factor and level.

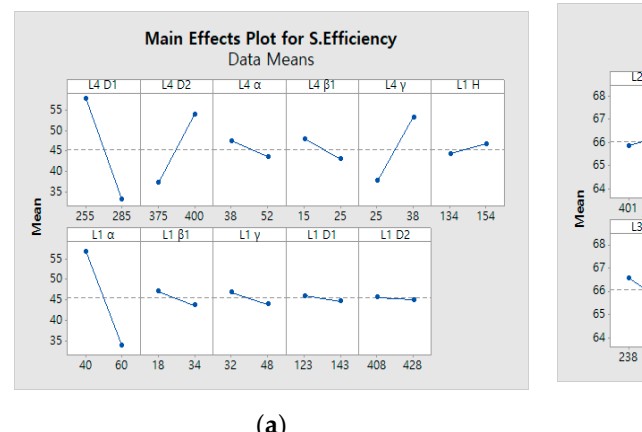
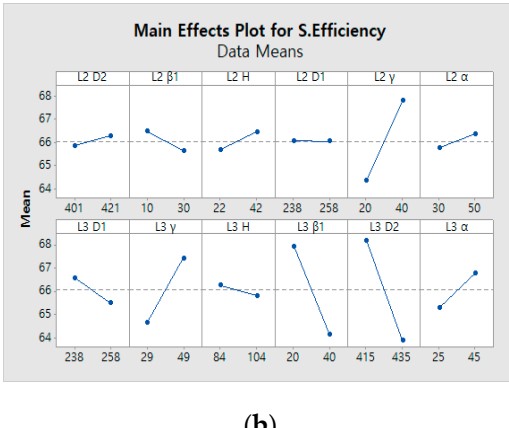

(**a**)                    (**b**)

**Figure 9.** Main effect plot of the airfoil parameters: (**a**) Layers 1 and 4; (**b**) Layers 2 and 3.

**Table 2.** Factors and levels.

| Layer | Factor | Level | | | Layer | Factor | Level | | |
|-------|--------|-------|---|---|-------|--------|-------|---|---|
| | | **−1** | **0** | **1** | | | **−1** | **0** | **1** |
| Layer 1 and 4 | $\alpha$ (Layer 1) | 20 | 35 | 50 | Layer 2 and 3 | $\gamma$ (Layer 2) | 20 | 35 | 50 |
| | $D_1$ (Layer 4) | 200 | 300 | 400 | | $\beta_1$ (Layer 3) | 5 | 15 | 30 |
| | $D_2$ (Layer 4) | 450 | 500 | 550 | | $D_2$ (Layer 3) | 450 | 500 | 550 |
| | $\gamma$ (Layer 4) | 20 | 35 | 50 | | $\gamma$ (Layer 3) | 20 | 35 | 50 |

After the first and second optimizations, the shape of the initial model could be determined. Once the initial shape was determined, the optimal design values were obtained from Layer 1 using the RSM. The iteration was performed by applying this value to the optimization of Layer 2. The calculations were repeated and finally converged to the target static pressure efficiency of 78% to complete the optimization. The optimal design values of the blades designed by the iteration method are shown in Table 3. The blade shapes of the original and optimized models are shown in Figure 10.

**Table 3.** Design values after blade optimization.

| Part | Parameters | Unit | Value | Part | Parameters | Unit | Value |
|------|-----------|------|-------|------|-----------|------|-------|
| Layer 1 | $D_1$ | mm | 340.0 | Layer 3 | $D_1$ | mm | 330.0 |
| | $D_2$ | mm | 475.0 | | $D_2$ | mm | 527.0 |
| | $\alpha$ | degrees | 42.5 | | $\alpha$ | degrees | 45.0 |
| | $\beta_1$ | degrees | 6.4 | | $\beta_1$ | degrees | 30.6 |
| | $\gamma$ | degrees | 45 | | $\gamma$ | degrees | 35.4 |
| Layer 2 | $D_1$ | mm | 298.0 | | $H$ | mm | 130.0 |
| | $D_2$ | mm | 444.0 | Layer 4 | $D_1$ | mm | 343.0 |
| | $\alpha$ | degrees | 47.5 | | $D_2$ | mm | 478.0 |
| | $\beta_1$ | degrees | 6.8 | | $\alpha$ | degrees | 45.0 |
| | $\gamma$ | degrees | 41.7 | | $\beta_1$ | degrees | 15.0 |
| | $H$ | mm | 42.0 | | $\gamma$ | degrees | 31.2 |

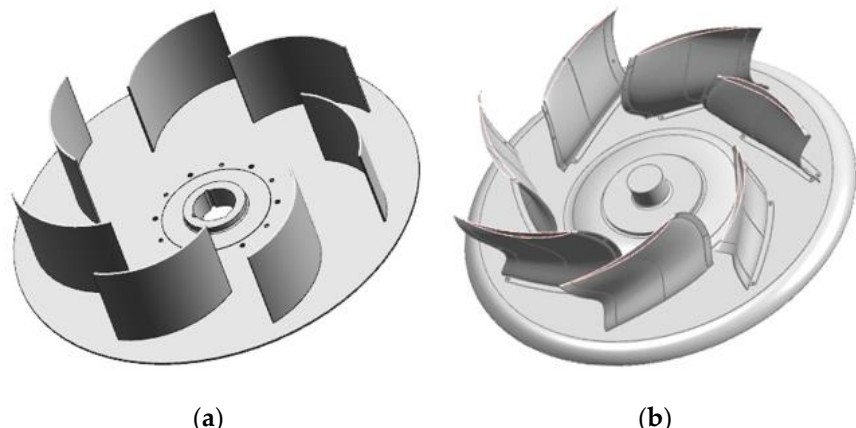

(a)             (b)

**Figure 10.** Geometric shape of the blades of the original and optimized models: (**a**) Original model; (**b**) Optimized model.

## 4. Numerical Results and Discussion

### 4.1. Mesh Independence Verification

Table 4 shows the mesh independence verification results of the optimized model. When the total elements were beyond 9,461,970, the change in static pressure was small.

**Table 4.** Mesh independence verification of the optimized model.

| No. | Mesh | Static Pressure (Pa) |
|-----|-----------|----------------------|
| 1 | 7,432,111 | 418 |
| 2 | 8,184,320 | 415 |
| 3 | 8,954,841 | 410 |
| 4 | 9,461,970 | 398 |
| 5 | 9,806,594 | 397 |

### 4.2. Comparison of the Performances of the Original and Optimized Models

Figure 11 shows the performance comparison between the original model and the optimized model. Both the static pressure and the static efficiency of the optimized model were improved compared with those of the original model. The highest efficiency of the optimized model was 78.1% at the point with a flow coefficient of 0.675, which was an improvement of more than 6.3% compared with the original model.

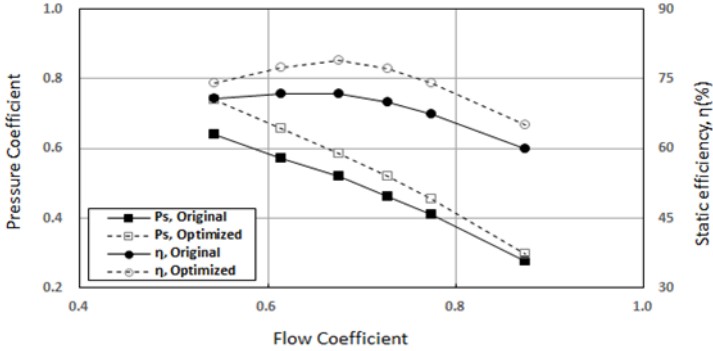

**Figure 11.** Comparison of performances between the original and optimized models (at 1100 rpm).

To verify the simulation results, two prototype fans made of ABS (Acrylonitrile Butadiene Styrene) were produced. Figure 12 shows an image of the prototype fan and the comparison between the simulation and experiment. It can be seen that the experimental results of the two prototype fans agreed well with the simulation results, within 3.1%.

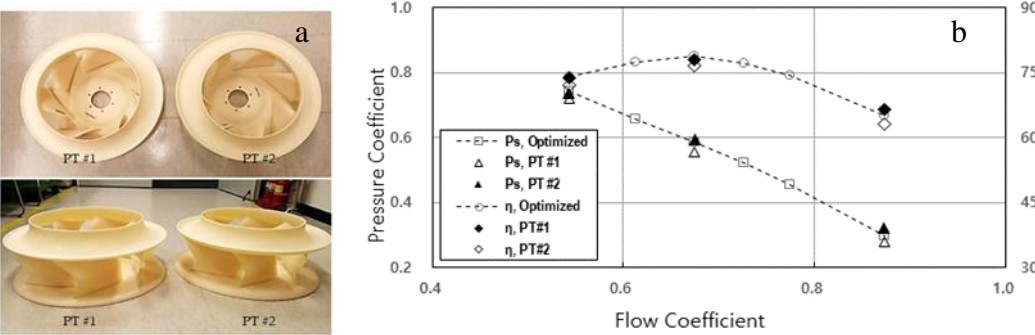

**Figure 12.** Prototype test results: (**a**) Prototype fan image; (**b**) Comparison of performance between simulation and experimental results of the prototype fan (at 1100 rpm).

Figure 13 shows the static pressure distribution in the middle section of the original and optimized models. In the optimized model, the static pressure in the channel increased uniformly and stably. By contrast, the original model showed a relatively large loss at the blade outlet.

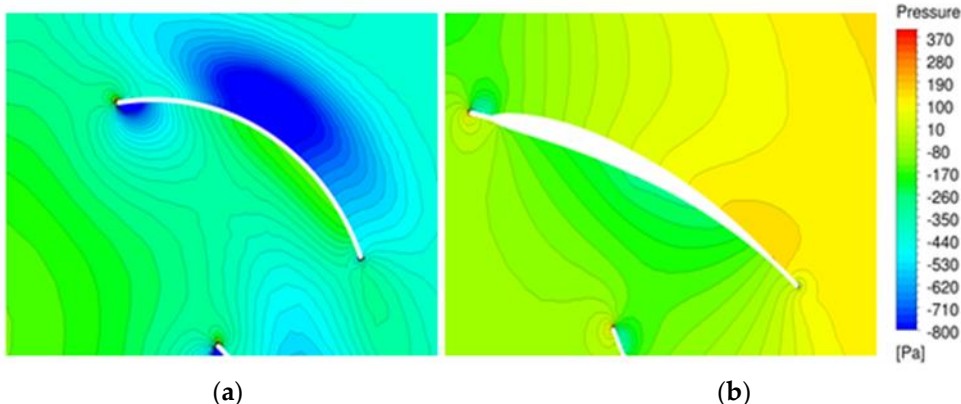

**Figure 13.** Static pressure distribution (at 1100 rpm): (**a**) Original model; (**b**) Optimized model.

Figure 14 shows the relative velocity distributions in the middle section of the original and optimized models. There was a low-velocity region on the suction surface of the original model, and there was a high-velocity region on the blade's outlet surface. In the optimized model, homogeneous velocity distributions on the suction surface and the channel could be observed. The inhomogeneous velocity distribution on the suction surface or outlet surface was the main cause of flow separation, which caused fluid flow friction, resulting in large losses and performance degradation.

Figure 15 shows the velocity streamline at the blade and the channel. In the original model, the fluid flow on the suction and outlet surfaces did not flow along the blade airfoil, and separation occurred. In the optimized model, on the other hand, some flow separation occurred at the blade tip, but the overall flow was stable along the blade airfoil. This stable flow reduced the fluid flow friction in the impeller to increase the performance and reduce noise.

Figure 16 shows the turbulence eddy frequencies of the original and optimized models. In the original model, turbulence increased because of flow separation from the leading edge. On the other hand, in the optimized model, the flow separation at the leading edge was significantly reduced, making the flow stable. Additionally, the loss in the impeller outlet was significantly reduced in the

optimized model. As a result, uniform and stable control of the flow in the blade can be seen as a major factor in increasing the efficiency of the plenum fan.

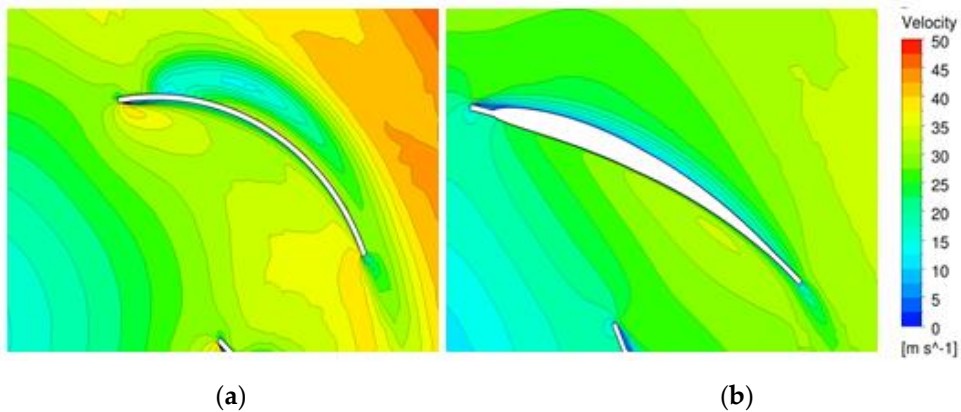

(**a**)          (**b**)

**Figure 14.** Velocity distribution (at 1100 rpm): (**a**) Original model; (**b**) Optimized model.

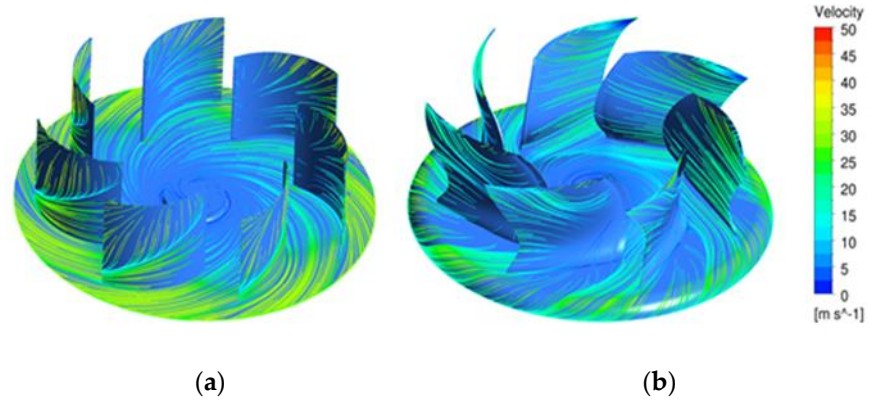

(**a**)          (**b**)

**Figure 15.** Velocity streamline on the surface (at 1100 rpm): (**a**) Original model, (**b**) Optimized model.

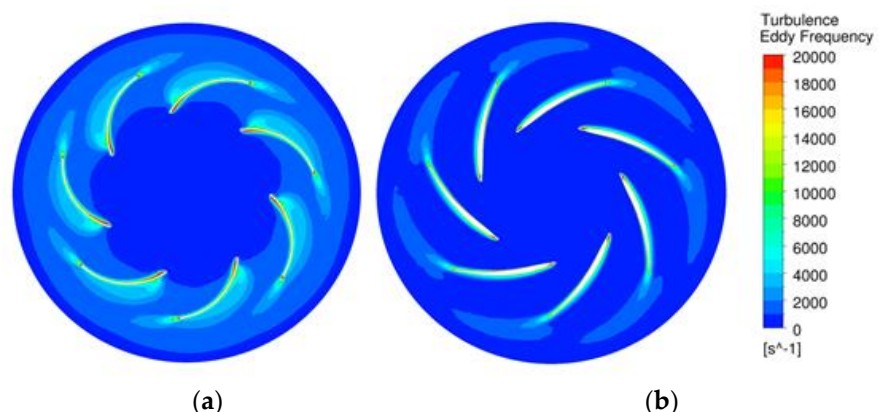

(**a**)          (**b**)

**Figure 16.** Turbulence eddy frequency (at 1100 rpm): (**a**) Original model; (**b**) Optimized model.

## 5. Conclusions

In this study, we optimized the performance of a plenum fan with a four-layer three-dimensional blade. The numerical calculations of the original and optimized models were based on the three-dimensional and incompressible RANS equation. The SST model, which is useful for the analysis of flow separation, was used for the turbulence model. The numerical results of the original model were in good agreement with the experimental results. The summarized conclusions are as follows.

1. The smooth, curved surface of the three-dimensional blade of the optimized model stabilized the flow and reduced the flow friction by restraining the flow separation as much as possible. The results show that both static pressure and static efficiency were improved in the optimized model compared with the original model. The static efficiency of the optimized model was improved by more than 6.3% compared with that of the original model at its peak.

2. In the original model, a relatively large loss occurred at the blade outlet. Additionally, in the original model, it was confirmed that turbulence grew because of flow separation from the leading edge. However, in the optimized model, the static pressure in the channel increased uniformly and stably. The flow separation at the leading edge was significantly reduced, which made the flow stable.

**Author Contributions:** Y.C.A. provided the basic idea for this study and administrated the project. K.J.L. provided numerical strategies, the basic concept of optimization, and worked on the analysis of numerical results. Y.M.K. carried out the numerical simulations. I.W.P. and K.S.B. carried out the experiments and comparison with simulation results. All authors have read and agreed to the published version of the manuscript.

**Funding:** This work was supported by the National Research Foundation of Korea (NRF) grant funded by the Korean government (MSIT) (No. 2018R1A2B6004137).

**Conflicts of Interest:** The authors declare no conflict of interest.

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
