# Peer review of "Optimal Design of a Plenum Fan with Three-Dimensional Blades"

_applsci, doi:10.3390/app10103460_

Round 1
Reviewer 1 Report
This paper shows an optimal design of a plenum fan with a three-dimensional curved blade. Although the results have demonstrated that the new design improve by 6% the original model; however, this paper seems to show a professional work rather than a research work. Moreover, the simulation shouldn't be considered for publication but if the authors could build the prototype and studying and comparing with the original model, this could be interesting.
Author Response
Reviewer(s)' Comments to Author:
Comments and Suggestions for Authors
This paper shows an optimal design of a plenum fan with a three-dimensional curved blade. Although the results have demonstrated that the new design improve by 6% the original model; however, this paper seems to show a professional work rather than a research work. Moreover, the simulation shouldn't be considered for publication but if the authors could build the prototype and studying and comparing with the original model, this could be interesting.
Answer:
We made two proto type fans made by ABS material to verify the simulation results.
The photo and the test result of the proto type fan are shown in Figure 12.
And test results and comparison results were added at line 249 to 252 of this paper.
We hope the manuscript is now acceptable for publication. Please let me know if you need anything else from us.
Thank you.

Reviewer 2 Report
The paper presents a study on optimized design of a plenum fan with three-dimensional blades. The reviewer has some questions for the authors regarding the manuscript content:
- Fig. 4, page 5 shows the comparison between the numerical and experimental results, could the authors provide more specific on the experimental test-rig, that is to say, the tested equipment, condition, test-rig picture?
- Fig. 7, page 6 shows the flow-chart of the iteration method of the blade optimization, could the authors provides the information on how long did it take to complete this full iteration for obtaining the final optimized model in Fig. 9(b).
- It seems to me that the optimized designed shows some advantages at the selected velocity (1100rpm). What happens if the velocity of the plenum fan is increased/reduced to another level? Could the authors provide comments on this scenario?
Author Response
Reviewer(s)' Comments to Author:
Comments and Suggestions for Authors
The paper presents a study on optimized design of a plenum fan with three-dimensional blades. The reviewer has some questions for the authors regarding the manuscript content:
Fig. 4, page 5 shows the comparison between the numerical and experimental results, could the authors provide more specific on the experimental test-rig, that is to say, the tested equipment, condition, test-rig picture?
Answer:
Added test equipment schematic diagram and test scene are shown in Figure 4.
Added additional explanation put in the paper at line 133 to 136.
Fig. 7, page 6 shows the flow-chart of the iteration method of the blade optimization, could the authors provides the information on how long did it take to complete this full iteration for obtaining the final optimized model in Fig. 9(b).
Answer:
Added computer specifications and convergence criteria at line 71 to 73.
(Calculations were performed on a Windows 10 based computer (Intel (R), 3.2 GHz, 128 core, 64 GB RAM) with a 64-bit operating system. For stability of analysis results, convergence criteria were set to 1x10-5.)
It took about 3 hours to simulate one case, and the time to complete the optimization was about 792 hours.
It seems to me that the optimized designed shows some advantages at the selected velocity (1100rpm). What happens if the velocity of the plenum fan is increased/reduced to another level? Could the authors provide comments on this scenario?
Answer:
In order to predict the fan performance when the operating conditions change, the fan affinity laws are generally applied.
(Fan affinity laws: The flow rate varies directly as the speed ratio, the pressure varies as the square of the speed ratio and power required varies as the cube of the speed ratio.)
If there is no change in the geometry of the blower and there is no change in the physical properties of the fluid being transported, fan affinity laws can be used to predict the performance of the fan according to the change in rotational speed.
This is thought to mean that the fluid flow inside the fan does not fluctuate dramatically due to the change in the number of revolutions of the blower.
In most previous studies, interpreting performance at rated speed is might be also the same reason.
We hope the manuscript is now acceptable for publication. Please let me know if you need anything else from us.
Thank you.

Reviewer 3 Report
The paper presents an improvement of a radial compressor using a screening method and response surface modeling. The methodology is quite clear with numerical and experimental results along with a comparison between optimized and baseline designs. The paper presents a rich analysis of the flow on the optimized design to identify the source of the improvement.
The novelty of the work is however not highlighted enough. The baseline design seems simplisitic. The optimization methodology is not well explained. The RSM method should be described with more details: training methodology, prediction accuracy, and contribution to the improvement.
Related work can help judge if 6% is a large improvement or not. As the baseline design is simplistic the 6% might be not very impressive.
the optimization considers only 1 operating point so the improvement may be at the cost of a reduction of the operating range. A multipoint optimization can be more general (unless the fan will be used mostly in the specified operating point)
Here is some detailed feedback:
line 60: there has been little research on improving the efficiency of
scroll-less plenum blowers -> It would help appreciate the 6% improvement reached in current work by citing similar work and their reached improvement.
line 85: y+ is a mesh quality measure. If not considered how can we judge if the results can be trusted?
line 107: the radius of the computational domain is 3 times greater than the fan diameter --> why do you opt for a large computational domain. Is it not possible to limit the domain to the fan and put its inlet and outlet as boundary conditions.
line136: optimum airfoil was designed for each layer --> a serial optimization of each layer does not take into account the interaction between layer. Why not optimize all layers at the same time using a Design of experiment for instance and a response surface?
line 150: please put reference for equ 6.
line 166: please put reference for the Taguchi method and explain it briefly as it is at the core of the current work.
line 176 : ¨The first optimization of Layers 1 and 4 was performed using the analysis of four factor-3 levels where selected as the main effect analysis using the response surface method (RSM).¨ --> please elaborate on the analysis of four factor and the role of the RSM.
line 180: what it L16 analysis
Author Response
Reviewer(s)' Comments to Author:
Comments and Suggestions for Authors
The paper presents an improvement of a radial compressor using a screening method and response surface modeling. The methodology is quite clear with numerical and experimental results along with a comparison between optimized and baseline designs. The paper presents a rich analysis of the flow on the optimized design to identify the source of the improvement.
The novelty of the work is however not highlighted enough. The baseline design seems simplisitic. The optimization methodology is not well explained. The RSM method should be described with more details: training methodology, prediction accuracy, and contribution to the improvement.
Related work can help judge if 6% is a large improvement or not. As the baseline design is simplistic the 6% might be not very impressive.
the optimization considers only 1 operating point so the improvement may be at the cost of a reduction of the operating range. A multipoint optimization can be more general (unless the fan will be used mostly in the specified operating point)
Here is some detailed feedback:
line 60: there has been little research on improving the efficiency of
scroll-less plenum blowers -> It would help appreciate the 6% improvement reached in current work by citing similar work and their reached improvement.
Answer:
Optimized model has improved efficiency by 6.3% point compared to original model. (Not 6%, but 6.3% point, the content of the paper has been modified.)
The improvement of 6.3% point from 71.8% to 78.1% may seem numerically small, but it can be said that there is a significant improvement with the highest efficiency at 78%.
Added new comment in the paper about previous studies as like below,
“Most of the previous studies focused on the improvement of the centrifugal fan with scroll, and in some scroll-less plenum fan studies, the airfoil had a simple 2D shape so the peak efficiency was relatively low at about 70%.”
line 85: y+ is a mesh quality measure. If not considered how can we judge if the results can be trusted?
Answer:
As you said, one of the biggest causes of errors in flow analysis is the improper use of turbulence models, and it is one of the most difficult tasks to fit y + to all areas, especially in the generation of a grid near a wall. In this study, the "automatic wall function" of CFX was used, and its goal was to secure the accuracy of the solution regardless of the possible y + of the grid.
While standard viscous sublayer models require a grid of y + ~ 1 to accurately analyze wall shear stress and wall heat transfer, the automatic wall function can also support very coarse wall grids. Since it is impossible to create a lattice with the same level of y + of all walls with a complex three-dimensional flow in a fluid machine, it was shown that this problem can be easily solved by using the "automatic wall function" of CFX.
line 107: the radius of the computational domain is 3 times greater than the fan diameter --> why do you opt for a large computational domain. Is it not possible to limit the domain to the fan and put its inlet and outlet as boundary conditions.
Answer:
Largely sized computational domains have the disadvantage of increasing computational loads.
However, in this study, the computational domain has been set as large as possible in order not to affect the operation of the fan for reasons such as recirculation among components of the suction flow or discharge flow based on the fan.
I think that we cannot simply use the fan area as a computational domain because we have to consider the flow of fluid at the same time in the simulation of the fan.
Previous studies have also set a large computational domain, such as holding the cylinder's diameter more than five times the fan's diameter.
line136: optimum airfoil was designed for each layer --> a serial optimization of each layer does not take into account the interaction between layer. Why not optimize all layers at the same time using a Design of experiment for instance and a response surface?
Answer:
In this study, iteration was performed by selecting the main factors for each layer of the airfoil and sequentially obtaining the optimum points of each layer to apply them to the calculation of the optimum value for the next layer.
While conducting research, It was found that sequentially optimizing for each layer and applying the optimum value of the previous section repeatedly to find the optimum value of the next section converged to a certain spec.
As you pointed out, The reason for not optimizing considering all factors at the same time for all layers is as follows.
1. The computing load when considering all factors is enormous.
2. This is because it is difficult to analyze and improve interactions between factors that occur when all factors of four layers are considered at the same time.
In the next study, I think I can consider how to perform optimizations at the same time, taking into account all the factors as your opinion.
line 150: please put reference for equ 6.
Answer:
In figure 7, the cord length is the opposite side of the angle alpha.
Connect the cord length and angle alpha, it becomes a triangle.
Here we can calculate the cord length using the second law of COS.
Equation 6 is a formula to calculate the cord length by applying the second law of COS.
Therefore, I think there is no need to put the references.
line 166: please put reference for the Taguchi method and explain it briefly as it is at the core of the current work.
Answer:
A reference to the Taguchi method has been added, and additional explanations have been written in the paper as like below;
Taguchi method[19,20] is used for initial factor selection. Taguchi method is a powerful statistical tool for designing an experiment utilizing orthogonal array tables. An orthogonal array that depends on the number of factors and levels are used to study the parameter space. Numerous input variables can be assessed to determine their contribution to the output response variable using a fewer number of experimental runs.[21] In this study, an experimental plan was established using the Taguchi method to select the initial factor values for each layer. Experimental plans were established using L12 (211) orthogonal arrays in layers 1 and 4, and using L16 (212) orthogonal arrays in layers 2 and 3.
line 176 : ¨The first optimization of Layers 1 and 4 was performed using the analysis of four factor-3 levels where selected as the main effect analysis using the response surface method (RSM).¨ --> please elaborate on the analysis of four factor and the role of the RSM.
Answer:
The description of the four factors selected in Layers 1 & 4, 2 & 3 was added to lines 202 to 214, and the factor and level for each layer were added to Table 2.
line 180: what it L16 analysis
Answer:
L16 means 12 factors, 2 level orthogonal array in the experiment plan of the 2nd and 3rd layer.
Corrected the text line 192 in the paper.
We hope the manuscript is now acceptable for publication. Please let me know if you need anything else from us.
Thank you.

Round 2
Reviewer 1 Report
This papers deserves to be published in the present form with the changes that authors have integrated.
Reviewer 3 Report
All comments have been addressed. For the optimization of the 4 sections simultaneously I suggest for a future work to use a DOE and any metamodel-assisted optimization method which can handle up to 50 different parameters.